# A Review of FLT3 Kinase Inhibitors in AML

**DOI:** 10.3390/jcm12206429

**Published:** 2023-10-10

**Authors:** Cristina Negotei, Andrei Colita, Iuliana Mitu, Anca Roxana Lupu, Mihai-Emilian Lapadat, Constanta Elena Popovici, Madalina Crainicu, Oana Stanca, Nicoleta Mariana Berbec

**Affiliations:** 1Department of Hematology, “Carol Davila” University of Medicine and Pharmacy, 050474 Bucharest, Romania; 2Clinic of Hematology, Coltea Clinical Hospital, 030171 Bucharest, Romania

**Keywords:** acute myeloid leukemia, FLT3 inhibitors, therapeutics, molecular biology, prognosis

## Abstract

Acute myeloid leukemia (AML) is a highly aggressive illness distinguished by the accumulation of abnormal hematopoietic precursors in both the bone marrow and peripheral blood. The prevalence of FLT3 gene mutations is high and escalates the probability of relapse and mortality. The survival rates for AML patients, particularly those over 65, are low. FLT3 mutation screening at diagnosis is mandatory, and FLT3 inhibitors are crucial in treating AML patients with mutations. There are two categories of FLT3 mutations: FLT3-ITD located in the juxtamembrane domain and FLT3-TKD in the tyrosine kinase domain. FLT3-ITD is the most common type, affecting nearly a quarter of patients, whereas FLT3-TKD only affects 6–8% of patients. FLT3 inhibitors are now crucial in treating AML patients with FLT3 mutations. When dealing with FLT3-mutated AML, the recommended course of treatment typically involves chemotherapy and midostaurin, followed by allogeneic hematopoietic cell transplantation (HCT) to maximize the likelihood of success. Maintenance therapy can lower the risk of relapse, and gilteritinib is a better option than salvage chemotherapy for relapsed or refractory cases. Clinical trials for new or combined therapies are the most effective approach. This review discusses treatment options for patients with FLT3-mutated AML, including induction chemotherapy and options for relapsed or refractory disease. Additional treatment options may become available as more studies are conducted based on the patient’s condition and susceptibility.

## 1. Introduction

Acute myeloid leukemia (AML) is the most aggressive myeloproliferative disorder, with a 5-year overall survival (OS) rate of 35% in the general population and below 10% in those above 65 years old [1]. AML pathogenesis is based on the accumulation of immature hematopoietic precursors by means of two main mechanisms: blocked differentiation and impaired intramedullary and extramedullary apoptosis [2]. AML is a complex disease with genetic and phenotypic differences. Researchers have successfully identified specific chromosomal and genetic variations that significantly impact the classification of AML as well as the associated risk level and appropriate course of treatment. These findings are a critical step towards improving patient outcomes and advancing the field of leukemia research [2]. FMS—like tyrosine kinase 3 (FLT3) gene mutations—are frequently detected in patients diagnosed with AML, with an incidence of 30%. Those with these mutations display a more increased chance of relapse and mortality than those without them. Consequently, FLT3 gene mutations represent the most prevalent cytogenetic abnormality in AML and pose a significant risk factor for unfavorable outcomes. To ensure optimal patient care, the World Health Organization strongly advises conducting FLT3 mutation screening for individuals diagnosed with AML [3]. There are two types of FLT3 mutations: internal tandem duplication (FLT3-ITD) within the juxtamembrane (JM) domain, which can be found in up to 25% of all patients, and the less common variant—tyrosine kinase domain (FLT3-TKD) mutation [3]. FLT3 ligand interacts with FLT3 receptor, leading to pathway activation and transcriptional changes. Type 1 inhibitors bind ATP sites, while type 2 inhibitors stabilize inactive conformation. TKD mutation prevents type 2 inhibitor binding and allows ATP binding [4].

This review aims to assess the therapeutic alternatives currently available for patients diagnosed with FLT3-mutated AML, including those with relapsed/refractory cancer. As studies progress, more treatment options may be available based on the patient’s condition and susceptibility. The primary objective of this review is to deliver a complete overview of the FLT3 inhibitors currently approved for managing AML with FLT3 mutations [1].

## 2. FLT3 Structure

FLT3 is a type III receptor tyrosine kinase crucial in cellular signaling. It is found on chromosome 13q12 and is known to include KIT, FMS, and PDGFR. This receptor is primarily described in hematopoietic stem cells, early myeloid, and lymphoid progenitor cells. Upon maturation into lymphoid or myeloid cells, the expression of FLT3 is either reduced or lost altogether. The receptor comprises five immunoglobulin-like domains in the extracellular region, a juxtamembrane (JM) domain, a tyrosine kinase domain (TKD) divided by a kinase insert domain, and a C-terminal domain in the intracellular region [2,5,6]. The structure of FLT3 is shown in Figure 1.

When the FLT3 ligand binds to the extracellular domain of the FLT3 receptors, it triggers dimerization with another FLT3 receptor. This process stimulates the intracellular kinase domains, leading to the phosphorylation of downstream proteins and the initiation of signaling cascades. These cascades regulate gene transcription, responsible for cell survival, proliferation, and differentiation [7].

**Figure 1 jcm-12-06429-f001:**
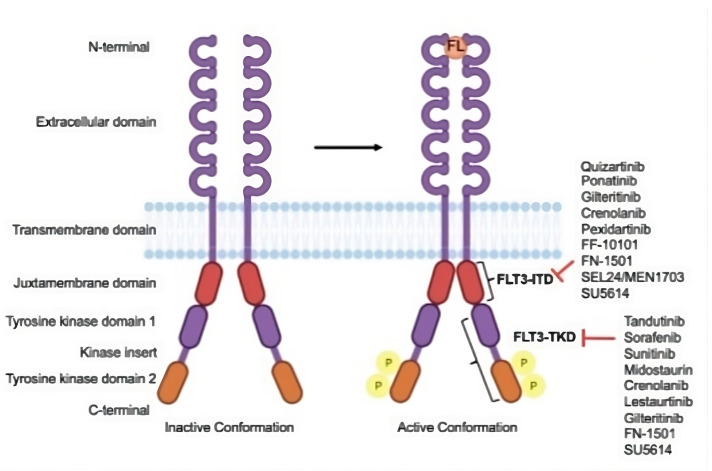
Structure of FLT3 and the drug targets. FLT3 is inactive in its conformation, but when it binds to FLT3 ligand (FL), it becomes active and undergoes autophosphorylation. There are various FLT3 inhibitors, each of which has a unique binding site on their domains. Adapted and reprinted from Ref. [8].

## 3. FLT3 Mutations

More than a thousand mutations of the FLT3 receptor have been described so far [9]. There are two major types of FLT3 mutations: (1) FLT3 internal tandem duplication (FLT3-ITD) in the juxta membrane (JM) domain of the protein and (2) point mutations in the FLT3 tyrosine kinase domain (FLT3-TKD) [10].

(1) FLT3-ITD occurs in 20–25% of AML adult patients and represents the most common mutation in AML. The ITD mutations are characterized by in-frame duplications with sizes ranging from 3 to 1236 pb that affect, in most cases, the JM region but also the beginning of the TKD [9].

(2) Point mutations in the FLT3-TKD are present in 7–10% of cases. The most described mutation is D835X, which is located in the activation loop of FLT3 and has various different substitution variants (D835Y, D835V, D835H, D835 E, D835N). Apart from this, other rarer mutations affecting FLT3 have been described: N676K, Y842C, N841I, N841K, K663Q, S451F, Y572C, V592G, R834Q, V592A, V579A, F594L, and F590GY591D. All these mutations cause constitutive FLT3 activation, resulting in cytokine-independent cell survival and cellular transformation [9].

## 4. Leukemogenesis in FLT3-Positive AML

FLT3 plays a key role: regulating normal hematopoiesis. It is a tyrosine kinase receptor involved early in the development of hematopoietic cells. It is typically expressed on both uncommitted precursors and common myeloid and lymphoid progenitors, with a single exception—the megakaryocyte-erythrocyte progenitor [11].

To understand the phenomenon of mutations in leukemia, it is crucial to conduct an extensive study of preleukemic hematopoietic stem cells (HSCs). In the case of AML, sequencing efforts have produced significant findings concerning mutation patterns. Notably, these endeavors have revealed that mutations such as DNMT3A, NPM1, and FLT3 frequently cluster together or are incompatible, such as TET2 and IDH1/2. These observations underline the complex interplay of genetic alterations in AML and highlight the importance of a thorough understanding of the mutational landscape in this disease. This evidence, therefore, supports the two-hit hypothesis of leukemogenesis, which states that AML appears as a result of at least two complementary mutations. Class I mutations represent an acquired proliferation advantage, and class II mutations impair cellular maturation and differentiation. Examples of class I mutations include FLT3-ITD, K-RAS, and KIT, while among class II mutations, the most common is CEBPA [12].

Regarding mutational behavior, there is an ongoing debate about whether FLT3 is a driver or passenger mutation. However, its role appears to be early in the pathogenesis of the disease [2].

The FLT3-ITD mutation is known to significantly impact the JM and TKD regions, which are responsible for maintaining the kinase autoinhibition process. As a result, the tyrosine kinase receptor is overexpressed or excessively activated, leading to ligand-independent FLT3 signaling and FL hyperresponsiveness, leading to downstream signaling pathways that preferentially stimulate myeloid cell expansion [8,13,14,15]. FLT3-ITD up-regulates MCL-1 to promote the survival of stem cells in acute myeloid leukemia via FLT3-ITD-specific STAT5 activation [16]. 

The conformational alterations induced by FLT3-ITD mutations in the JM region lead to permanent activation of TKD, resulting in abnormal proliferation and differentiation of myeloid progenitors. Signaling induced by FLT3-ITD differs from that of wild-type FLT3 and implies enhanced STAT5 phosphorylation that leads to upregulation of BCL-X and PIM1 involved in the inhibition of apoptosis [17].

Several factors, including the allelic ratio of the mutation, the presence of co-mutations, and the insertion site, influence the outcome for patients with FLT3-ITD mutation [18]. Isolated FLT3-ITD mutation cannot be responsible for the occurrence of acute myeloblastic leukemia, and it must be associated with other molecular abnormalities: NPM1 mutation, AML1-ETO gene fusion, NUP98 fusion, CBFβ-SMMHC fusion gene, and TET 2 deletion [19].

The presence of ITD mutations in the FLT3 gene is related to raised blast counts and leukocytosis. It also indicates an adverse prognosis for overall survival (OS), relapse-free survival (RFS), and event-free survival (EFS) [7]. As for the FLT3-TKD mutation, the associations are less clear and more dependent on the other cytogenetic and molecular aberrations, and the presence of leukocytosis may be observed; however, its prominence is not as significant as in instances with ITD mutations [7].

Point mutations in FLT3-TKD induce continuous, FL-independent activation of FLT3. The proliferative signaling cascades involved in leukemogenesis are different from signaling induced by FLT3-ITD. FLT3-TKD does not activate the JAK/STAT5 pathway but exerts inhibitory effects on JAK signaling through enhanced SHP1 and SHP2 activity. This could be a possible explanation for the different, less aggressive phenotype of FLT3-TKD compared to FLT3-ITD [17].

Studies have shown that more than half of patients with de novo AML and FLT3 mutation exhibit the same mutation upon relapse. Nevertheless, with the advent of FLT3 inhibitors, it is anticipated that this percentage will decrease. This noteworthy occurrence of FLT3 mutation during relapse is a sign that the FLT3-ITD clone, which was present during the initial diagnosis, can expand and cause a recurrence of AML [20].

## 5. Previous Treatment in FLT3+ AML

AML does not represent a single disease but an entire group of diseases, each with specific underlying characteristics and unique prognoses. The “3 + 7” regimen (three days of daunorubicin and seven days of cytarabine) was conceived in the late 1970s and has become a widely accepted standard of care. Survival rates using this regimen remain low: only 1 in 3 young patients (those under 60) and 1 in 10 old patients (those aged 60 or older) achieve the 5-year cut-off [21]. Considerable advancements in survival rates for eligible patients with intermediate- or high-risk AML have been made over time by using HCT [22].

The typical age range for patients diagnosed with AML falls within the median of 68 to 70 years. However, it is important to note that older patients are often excluded from trials focusing on intensive chemotherapy (IC) treatments, such as the 3 + 7 regimen, as IC requires a high fitness level. A study was conducted on a cohort of 813 patients aged over 60 years (with a median age of 67) who received a 3 + 7 regimen. The study revealed that the average survival rate was between 7 and 8 months, with an estimated 20% 3-year survival. The Surveillance, Epidemiology and End Result database examined data from approximately 29,000 patients diagnosed with AML. This study presents a more accurate representation of the situation. In nearly two decades, the results observed among patients undergoing recent treatment failed to meet expectations. Patients with de novo AML (excluding APL and CBF-AML) aged 70 years or older had a 50% four-week mortality rate and less than 5% 5-year survival rate. Patients in the younger group had a 27% early mortality rate and a 40% 5-year survival rate [21].

In the past, treatment options for AML patients who were elderly or physically unfit were limited. Minimal advantages were provided through the use of DNA methyltransferase inhibitors (DNMTIs, i.e., azacitidine and decitabine) and low-dose cytarabine (LDAC). Venetoclax has been approved by the United States Food and Drug Administration (FDA) for use in patients aged 75 or older in conjunction with DNMTi or LDAC. The efficacy of venetoclax in combination with DNMTi was evaluated in an international, multicenter, randomized, double-blind, placebo-controlled, phase III study, VIALE-A, involving 431 patients, which resulted in a 36.7% complete remission with azacytidine-venetoclax versus 17.9% with Azacitidina-placebo [23]. 

Previously, AML treatment was considered a “one-size-fits-all” approach due to the lack of benefits from personalized induction therapy compared to fixed-schedule induction therapy. Starting with 2017, there have been 11 new medications and regimens authorized by the FDA, with 8 of these especially designed to target particular subgroups. In order to ensure the successful personalization of treatment plans, it is necessary to carefully evaluate a patient’s suitability for intensive chemotherapy and monitor any changes in clonal genetic heterogeneity that may occur through treatment, in addition to analyzing chromosomal and genetic information [24,25].

Regarding the patients diagnosed with AML FLT3-ITD, based on the EORTC GINEMA AML-12 trial, it has been observed that OS, regardless of age, was higher on the HD cytarabine arm compared to SD cytarabine (younger/older: HR = 0.7/0.8; *p* = 0.02/0.14) [26]. 

The detection of FLT3 mutations has been a subject of interest for over two decades, and the clinical application of FLT3 inhibitors has since become available for treating AML patients with such mutations. Despite overcoming several challenges in the development process of FLT3 inhibitors, it is imperative to conduct further in-depth investigations to identify the optimal treatment protocols for individual patients [27].

## 6. FLT3 Inhibitors

Medical research has witnessed a significant breakthrough in producing FLT3 inhibitors in recent years, leading to their emergence in clinical trials. These inhibitors are intended for use in single-agent or combination therapy, and they are currently being evaluated for their effectiveness in the treatment of various stages of diseases. Ref. [28] FLT3 tyrosine kinase inhibitors are a diverse group of drugs that showcase a wide range of variations in their effectiveness, preference, binding capacity, and site. Type I inhibitors, such as midostaurin, gilteritinib, and crenolanib, bind to the kinase-active conformation of FLT3, while type II inhibitors, including quizartinib and sorafenib, bind to the inactive structure [1]. Table 1 provides a thorough examination of the first-generation FLT3 inhibitors that are currently being implemented in clinical trials for AML therapy.

Based on their receptor specificity, FLT3 inhibitors can be categorized into two groups, namely, first generation or second generation. The former, including sunitinib, sorafenib, midostaurin, and lestaurtinib, exhibit an insufficiency of specificity and can impI ede multiple receptor tyrosine kinases and pathways, thus providing a wide spectrum of efficacy in AML patients. Second-generation inhibitors possess FLT3 receptor specificity, which hypothetically translates into fewer therapy-related toxicities and side effects. Quizartinib, crenolanib, and gilteritinib are representatives of this latter group [8]. 

FLT3 inhibitors, such as Midostaurin, Sorafenib, Gilteritinib, Quizartinib, and Crenolanib, have been known to cause serious adverse effects. The most commonly reported are hematologic and constitutional in nature. Patients may experience pancytopenias, differentiation syndrome, cardiac ischemia, abdominal discomfort, diarrhea, peripheral neuropathy, headache, and flu-like symptoms. These side effects can be severe in approximately 5–6% of patients. It is worth noting that both types of FLT3 inhibitors can cause such side effects [20,29].

Hematological toxicity is more prevalent with type 1 inhibitors (74.6%, 95% CI 70–79%), followed by gastrointestinal symptoms (22.1%, 95% CI 19–25.4%) such as dyspepsia, diarrhea, or pancreatitis. Type II inhibitors still cause hematological toxicity in 57.7% of cases (95% CI 54.6–60.8%) and GI manifestations in 13.9% of patients (95% CI 12–16%). Cardiac toxicity, specifically prolongation of QTc, is another potentially life-threatening side effect. Type 1 inhibitors have a 2% incidence of QTc prolongation (95% CI 1–3.6%), with gilteritinib being the most common molecule involved. Type 2 inhibitors have a 7% incidence of QTc prolongation (95% CI 5.3–9%), with quizartinib causing dose-dependent toxicity that can be severe. It is important to keep these potential side effects in mind when considering FLT3 inhibitors as a treatment option [20,29].

**Table 1 jcm-12-06429-t001:** A comprehensive overview of first-generation FLT3 inhibitors utilized in clinical trials for AML treatment.

	Drug	Targeted Mutation	Study (Ref.)	Usage	Benefit
Front-line therapy	Midostaurin	FLT3-ITD/TKD	RATIFY (phase III) [30]	Standard of care “3 + 7” + Midostaurin	OS: 74.7 vs. 25.6 mo
		FLT3	-	In combination with Azacitidine (not eligible for IC)	OS: 22 wks
	Sorafenib	FLT3-ITD	NCT02196857 (phase II) and NCT01254890 (phase I/II) [31]	In combination with Azacitidine (not eligible for IC)	OS: 8.3 mo
		FLT3	SORAML (phase II) [32]	Standard of care “3 + 7” + Sorafenib	EFS: 21 vs. 9 mo
	Lestauritinib	FLT3	-	Various intensive TX	OS: 46% vs. 45%
Relapsed/refractory disease	Midostaurin	FLT3	NCT00045942 (phase II b) [33]	Monotherapy	OS: 130 d
Maintenance post-HCT	Sorafenib	FLT3-ITD	SORMAIN (phase II) [34]	Monotherapy	55 OS: NR
	Midostaurin	FLT3-ITD	RADIUS (phase II) [35]	Monotherapy	22 OS: NR

Abbreviations: FLT3, FMS- like tyrosine kinase; ITD, internal tandem duplication; TKD, tyrosine kinase domain; HCT, hematopoietic cell transplantation; OS, overall survival; IC, intensive chemotherapy; mo, months; NR, not reached.

### 6.1. Midostaurin

Midostaurin is a first-generation inhibitor that is highly effective and explicitly targets ITD and TKD mutations. Its inhibitory effects extend to various protein kinases including c-KIT, VEGFR, and PDGFR-β. This therapeutic agent has been established as a highly adequate treatment option for patients with relapsed or refractory acute myeloid leukemia. Midostaurin has successfully reduced or eliminated leukemic blasts in both FLT3-mutated and FLT3-WT patients. In 2017, the FDA granted approval for the use of Midostaurin in treating AML patients with FLT3 mutations. Recently, it has been accepted for use in newly diagnosed patients with the FLT3 mutation or systemic mastocytosis [6].

The effectiveness of Midostaurin in combination with standard therapy for treating FLT3 (ITD or TKD) AML in patients under 60 years old was studied in the randomized phase III RATIFY trial. The analysis concluded that Midostaurin, when used with 3 + 7 induction and high-dose Cytarabine consolidation, is now the standard of care for newly diagnosed FLT3mut AML patients. This treatment was authorized by the FDA, EMA, and AIFA based on the trial results. However, Midostaurin did not prove valuable in the maintenance setting in the RATIFY or RADIUS trials [36]. 

For youths with FLT3 ITD-mutated AML who had undergone HCT, the phase II AMLSG 16–10 trial researched the efficiency of midostaurin as a maintenance treatment. In the course of the study, patients who underwent transplantation were administered a dose of 50 mg of midostaurin twice a day for a year. The outcomes reveal that the treatment led to a superior 1-year EFS contrasted to previous cases involving AML patients with FLT3 ITD mutations [37].

### 6.2. Sorafenib

Sorafenib is classified as a type II FLT3 inhibitor of the first generation. Despite its potential benefits, it has yet to receive approval for treating AML [1]. During the SORAML phase II study, sorafenib was provided to patients aged 60 years or younger with FLT3 AML at diagnosis. Although it demonstrated notable improvement over placebo in the 5-year EFS (41% vs. 27%; *p* = 0.011) and the RFS (53% vs. 36%; *p* = 0.035), sorafenib failed to improve the 5-year OS (61 vs. 53%; *p* = 0.282). The statistical analysis did not reveal any substantial differences in the response rate, disease-free survival (DFS), or OS between patients with FLT3-ITD and those without this mutation. A randomized phase II study (ALLG AMLM16) was conducted between Jan 2013 and May 2018 to determine whether sorafenib combined with intensive chemotherapy in newly diagnosed FLT3-ITD-mutated AML shows any survival benefit. The addition of sorafenib to intensive chemotherapy did not increase EFS. It might be worth mentioning that OS did improve with sorafenib among patients with higher FLT3-ITD allelic ratio or receiving an HCT after the first complete remission. Unfortunately, the study lacked the statistical capacity to determine substantial results [38]. 

Post-transplant maintenance therapy for patients with FLT3-ITD-mutated AML has been a topic of interest in recent years. Sorafenib has been considered a potential option for this population. A study conducted for a median period of 42 months demonstrated that sorafenib significantly decreases the risk of relapse or death compared to placebo. Using sorafenib maintenance therapy resulted in an 85% RFS rate and a 90.5% OS rate, whereas the placebo arm reported an RFS rate of 53.3% and an OS+ rate of 66.2% [24].

### 6.3. Sunitinib, Lestaurtinib, Tandutinib

Other FLT3 inhibitors of the first generation have shown insufficient antileukemic effects when used alone and inconsistent outcomes when used in conjunction with chemotherapy. For patients with recurrent or refractory disease, sunitinib, lestaurtinib, and tandutinib have all shown limited and brief responses when used as a single-agent therapy. A phase I/II study conducted on a cohort of individuals over the age of 60 found that sunitinib at once with first-line chemotherapy provided a complete response rate of 59%. However, it is essential to note that some patients experienced toxicities due to this treatment regimen. In a recent phase III trial, patients diagnosed with FLT3-mutated AML were randomly assigned to receive lestaurtinib in conjunction with induction and consolidation chemotherapy. Despite this approach, there was no notable difference in the main outcomes of overall survival and relapse-free survival between the lestaurtinib group (with a 5-year OS of 46% and a 5-year RFS of 40%) and the control group (with a 5-year OS of 45% and a 5-year RFS of 36%). In a randomized phase III clinical trial, patients with relapsed FLT3-mutated AML were set to receive chemotherapy with or without the administration of lestaurtinib. The trial found no statistical difference in complete remission rates between the two groups, with a 26% rate for chemotherapy-only and a 21% rate for chemotherapy and lestaurtinib. Currently, the elaboration of these AML inhibitors has ceased, and they have yet to receive approval for utilization [29].

### 6.4. Giltertinib

Gilteritinib Quizartinib and Crenolanib are representatives of FLT-3 second-generation inhibitors. Table 2 provides a comprehensive analysis of the second-generation FLT3 inhibitors that are presently undergoing clinical trials for AML treatment.

Gilteritinib is a next-generation drug that specifically targets two types of receptors, FLT3 and AXL (oncogenic tyrosine kinase), by inhibiting the activity of tyrosine kinase. Its ability to impede AXL is particularly beneficial, as AXL activation is a well-known mechanism of resistance to FLT3 inhibitor. By inhibiting AXL, Gilteritinib can effectively slow down the growth of FLT3-ITD AML tumors. This makes it a highly powerful and selective inhibitor with great potential for treating various types of cancer [45].

The phase III ADMIRAL trial conducted a comparative study between gilteritinib and salvage chemotherapy for 371 patients who were diagnosed with relapsed or refractory FLT3-mutated AML. The gilteritinib group gave them better outcomes in terms of median OS (9.3 vs. 5.6 months), CR/CRi percentage (34% vs. 15.3%), and EFS duration (2.8 vs. 0.7 months). These findings suggest that gilteritinib might be a better option for patients with R/R FLT3-mutated AML than intensive chemotherapy [38].

According to the 2-year follow-up of the ADMIRAL trial, it was found that gilteritinib demonstrated long-term survival advantage in individuals with FLT3-ITD mutation and those with high FLT3-ITD allelic ratio. However, it was not observed to provide the same benefit to those who belonged to the FLT3-TKD subclass or patients with a low FLT3-ITD allelic ratio. These findings suggest that gilteritinib may be a useful treatment option for certain types of patients with FLT3-ITD mutations. Those treated with this inhibitor who reached full response or combined CR had high relapse rates after 2 years (52.6% and 75.7%, accordingly) [45].

Gilteritinib in combination with HSCT could be a promising option for patients with R/R AML, as it boasts a higher response rate and reduced toxicity than intensive SC. This approach has the potential to make the regimen more bearable and decrease complications associated with transplantation. Furthermore, the usage of gilteritinib maintenance therapy has shown promise in prolonging remission after hematopoietic stem cell transplantation [46].

### 6.5. Quizartinib

Single-agent quizartinib has been shown to be highly effective in clinical trials for patients with relapsed/refractory AML, targeting FLT3 as a second-generation TKI. When compared to its first-generation counterparts, it exhibits a significantly more selective action against FLT3. Research has shown that the development of resistance to quizartinib in individuals with AML is mainly correlated with mutations in the FLT3 gene. Conversely, resistance to gilteritinib is frequently observed in approximately 60% of AML patients due to mutations in numerous ligands [47].

In a randomized phase 3 trial (QuANTUM-First), patients with newly diagnosed FLT3-ITD AML received Quizartinib as part of their treatment, which included standard anthracycline and cytarabine-based induction and consolidation, with quizartinib taken at a dose of 40 mg daily on specific days. The trial found that the inclusion of Quizartinib led to a subtle increase in complete remission or complete remission with poor hematologic recovery rates, with respective percentages of 71.6% and 64.9%, when compared to the placebo. Furthermore, it was found that the administration of Quizartinib resulted in a statistically considerable improvement in overall survival, with a median overall survival of 31.9 months in the Quizartinib group, compared to 15.1 months in the placebo group. Also, Quizartinib demonstrates a longer median duration of CR at 38.6 months (95% CI: 21.9, NE) compared to placebo at 12.4 months (95% CI: 8.8, 22.7); these results led to FDA approval in July 2023 for patients newly diagnosed with FLT3-ITD AML [24,48]. 

The clinical benefit of Quizartinib administered in monotherapy to patients with relapsed or refractory FLT3-ITDmut AML was investigated in a randomized phase IIb study and was found to be 47%. This promising outcome led to the initiation of a randomized phase III trial, QuANTUM-R, which aimed to compare Quizartinib in single-agent administration with researcher selection salvage chemotherapy in a cohort of 367 patients. The trial showed that Quizartinib had a considerable effect on the overall survival time, surpassing the outcomes of salvage chemotherapy by 1.5 months [36].

Quizartinib was first authorized in Japan in 2019 for use in the treatment of relapsed or refractory AML cases with FLT3-ITD mutation. A phase I trial was conducted to investigate the effectiveness of quizartinib in decreasing relapse rates for FLT3-ITD AML patients post-HSCT [36]. In a detailed analysis of a subgroup comprising 89 patients out of 208 who were administered with maintenance therapy utilizing either quizartinib or a placebo after consolidation chemotherapy, recently published results from a QuANTUM-First study showed the OS HR was 0.4 (95% CI: 0.19, 0.84). Another subgroup analysis consisting of 119 patients out of 208 who received maintenance therapy with quizartinib or placebo following HSCT showed an OS HR of 1.62 (95% CI: 0.62, 4.22). Although the CR rate in the quizartinib arm and the placebo arm is the same (55% with 95% CI: 48.7, 60.9 for quizartinib vs 95% CI: 49.2, 61.4 for placebo), there are significant differences in the median duration of CR [48].

### 6.6. Crenolanib

Crenolanib is a second-generation FLT3 inhibitor that exhibits selective suppression of both wild-type and mutant forms of FLT3. Originally, it was created as a platelet-derived growth factor receptor (PDGFR) inhibitor. This specific therapy has shown promising results in preclinical research, indicating a significant decrease in bone marrow depression compared to quizartinib. The lower impact on the proliferation of bone marrow cell colonies is a crucial factor contributing to this outcome [38].

In a phase II trial, crenolanib was assessed for its safeness and effectiveness in treating patients recently diagnosed with FLT3 AML. The outcomes of the study were encouraging, with an overall response rate (ORR) of 96%, which included a CR rate of 88%. Moreover, the OS rate was over 80%, which was observed during a median follow-up period of 6.2 months. These results signify that crenolanib has the potential to become a promising treatment option for FLT3-mutated AML patients [6].

## 7. Resistance to FLT3 Inhibitors

### 7.1. Primary Resistance

Primary resistance to FLT3 inhibitors could be explained by various mechanisms: FL bypassing, FLT3-independent MAPK activation, microenvironment-mediated mechanisms: cell adhesion, cytokine, growth factors (e.g., fibroblast growth factor 2 (FGF2)), soluble proteins, and degradation of FLT3 inhibitors. (e.g., CYP3A4 expression in medular stromal cells) [17,27].

### 7.2. Secondary Resistance

Secondary resistance against FLT3 inhibitors is caused by on-target and off-target mechanisms.

#### 7.2.1. On-Target Resistance 

On-target resistance is characterized by resistance to FLT3 inhibitors through acquired mutations in the FLT3 gene while leukemic cells are still dependent on FLT3 signals. The first mutations were described in patients with FLT3-ITD relapsing after quizartinib therapy. Mutations in TKD localized in the D385 residue, Y842 residue, or F691 gatekeeper residue confer resistance to the antileukemic effects of type II FLT3 inhibitors. Molecular analysis suggests the presence of more than one TKD mutation or association of several mechanisms in resistant patients [27].

#### 7.2.2. In Off-Target Mechanisms

In off-target mechanisms, leukemia cells become dependent on other signaling pathways. This category of mechanism was described mainly in patients treated with type I inhibitors (gilteritinib, crenolatinib) and consists in additionally acquired FLT3 mutations or acquired mutations affecting alternative pathways (RAS/MAPK) or affecting epigenetic regulators, myeloid transcription factors, and the cohesin complex in FLT3 mutation-independent clones [17,27].

## 8. Further Directions

When approaching the treatment of a newly diagnosed patient with FLT3-mutated AML, the eligibility for an allogeneic transplant of HSC plays a crucial role in determining the most effective approach. In the event that the patient is fit, young, without significant comorbidities, and eligible for Allo HCT, the standard course of treatment involves induction chemotherapy with a first-generation FLT3 inhibitor (midostaurin or sorafenib), followed by high-dose cytarabine consolidation (HiDAC) and Allo HCT if complete remission is achieved. If Allo HCT is not a viable option, single-agent FLT3 inhibitor maintenance is continued for a duration of 12 months. In the event of relapse during treatment, the second line of treatment involves a second-generation FLT3 inhibitor in monotherapy (gilteritinib), which is continued for up to 2 years as maintenance therapy, even if Allo HCT has been performed [14,49].

For those patients who are deemed unfit for Allo HCT, the standard therapy involves the combination of venetoclax and HMA. However, a more effective therapeutic approach would be a triple therapy strategy involving the combination of the aforementioned double therapy with an FTL3 inhibitor [14,49].

The necessity of triple therapy for elderly patients is reliant on the efficacy and safety outcomes of previously studied combinations. The randomized, open-label, phase 3 LACEWING study showed no benefits for the combination of azacitidine + gilteritinib compared to azacitidine alone. Venetoclax, in association with tyrosine kinase inhibitors, is a better combination, considering the two molecules work together to facilitate apoptosis. However, although the synergistic effect of Venetoclax with gilteritinib has been demonstrated, this combination also presents several tolerability issues. There is ongoing research regarding the triple association—hypomethylating agent, venetoclax, and FLT3 inhibitor—with anticipated results regarding efficacy and toxicities in the unfit or ASCT-ineligible AML-FLT3-positive patient [14].

## 9. Conclusions

AML with FLT3 mutations is a prevalent, complicated, and high-risk disease that requires a comprehensive understanding of cellular signaling, microenvironment interactions, and the immune system’s function. FLT3 inhibitors are now a fundamental part of treating patients with FLT3-mutated AML. Nevertheless, with the constantly evolving medical practices, several challenges remain to be addressed. 

AML is a heterogeneous disease, posing a constant threat, as it leads to additional mutations when new therapies are used. One effective strategy to address mutations is to undertake clinical trials that incorporate novel or combined therapeutic approaches while simultaneously investigating the mechanisms of resistance that emerge during relapse. This approach can provide valuable insights into the resistance mechanisms and inform the development of more effective treatments for patients.

For patients newly diagnosed with FLT3-mutated AML, the current treatment involves a combination of 3 + 7 induction chemotherapy and midostaurin. To lower the risk of relapse, allogeneic HCT is often advocated as a post-remission treatment. While the use of FLT3 inhibitors for post-transplant maintenance has yet to be established, experts suggest that maintenance therapy could help lower the risk of relapse. Studies have shown that Gilteritinib is a superior treatment option compared to salvage chemotherapy for patients with relapsed or refractory FLT3-mutated AML. Further clinical studies are ongoing and are expected to offer more treatment options based on patients’ vulnerability to FLT3-mutated AML and the stage of the disease.

## Figures and Tables

**Table 2 jcm-12-06429-t002:** A comprehensive overview of second-generation FLT3 inhibitors utilized in clinical trials for AML treatment.

	Drug	Targeted Mutation	Study (Ref.)	Usage	Benefit
Front-line therapy	Crenolanib	FLT3-ITD/TKD	Phase II study [39]	Standard of care “3 + 7” + Crenolanib	OS: 88%
	Gilteritinib	FLT3-ITD/TKD	NCT02236013 (phase I) [40]	Standard of care “3 + 7” + Gilteritinib	OS: 35.8 mo
	Gilteritinib	FLT3-ITD/TKD	Lacewing (phase III) [41]	In combination with Azacitidine (not eligible for IC)	OS: 9.8 vs. 8.9 mo
	Quizartinib	FLT3-ITD	QuANTUM-First (phase III) [42]	Standard of care “3 + 7” + Quizartinib	OS: 31.9 vs. 15.1 mo
Relapsed/refractory Disease	Gilteritinib	FLT3-ITD/TKD	Admiral (phase III) [43]	Monotherapy	OS: 9.3 vs. 5.6 mo
	Quizartinib	FLT3-ITD	QuANTUM-R (phase III) [44]	Monotherapy	OS: 6.2 vs. 4.7 mo

Abbreviations: FLT3, FMS-like tyrosine kinase; ITD, internal tandem duplication; TKD, tyrosine kinase domain; HCT, hematopoietic cell transplantation; OS, overall survival; IC, intensive chemotherapy; mo, months.

## Data Availability

Not applicable.

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
