# Peer review of "A Review of FLT3 Kinase Inhibitors in AML"

_jcm, 2023, doi:10.3390/jcm12206429_

Round 1

Reviewer 1 Report

Cristina Negotei and coll express in their review about FLT3 kinase inhibitors in AML clear concepts on their use in clinical practice. Clinical studies are clearly described and the potential usefulness of TKIs in AML field is well supported.

It could be useful describe a sort of "ideal" algorithm for the correct use of TKIs inhibitors in AML pre and post allogeneic transplant to complete the review

Reviewer 2 Report

This is a review of FLT3 mutated AML. 

Comments:

1.  In the introduction, would mention the different receptor types as this can determine response to various inhibitors. . 

2. Was permission obtained to reprint Figure 1?  also, some of the inhibitor placements look incorrect; e.g., quizartinib can result in emergence of TKD mutations. 

3. LIne 88; leukocytosis can be seen but just isn't as prominent as with ITD mutations. 

4 .Lines 97 and 98; would make this clearer; some mutations can commonly be seen together and some are mutually exclusive. 

5 .Line 107, If FLT3 is an early driver mutation, why  is it often lost at relapse; in other words, it comes and goes. 

6. In section 5, would indicate how those with FLT3 mutations fared with these standard therapies. 

7. Venetoclax is approved for those 75 and above; not 60. 

8. In Table 2, Is NR not reached? Each line/study should be associated with a reference .

9.Update quizartinib FDA approvals and recent studies related to that. 

10. It would be helpful to the reader to indicate how these inhibitors differ in their side effects .

11.  Since most studies have only looked at patients under 60, would discuss options for treating elderly patients with FLT3 mutated disease. 

Generally good but some minor editing needed. 

Reviewer 3 Report

The authors give an overview of FLT3 inhibition in acute myeloid leukemia. The review is well organized, simple to be received, and well written. However, it is pretty short in the context and requires further expansion of the described studies. My suggestions aim to enlarge the backbone of the review keeping in mind that the topic is discussing pharmacologic agents targeting FLT3 mutants:

-Expand the part on the pathogenesis of FLT3 mutant acute myeloid leukemia.

-Describe the molecular profile of FLT3 mutants.

-Describe mechanisms of resistance.

Round 2

Reviewer 2 Report

The content of the manuscript is improved. 

Do you have permission to reprint Figure 1?

Line 123--led should be lead

LIne 191--remove the word positive. 

LInes 384-6--this seems to be misplaced and would be better in the 2nd paragraph where upfront therapy is discussed. 

Line 450--what is LAM? 

Some minor style editing required. 
